# Methodological variations affect the release of VEGF *in vitro* and fibrinolysis' time from platelet concentrates

**Leonel Alves de Oliveira**[1], **Tatiana Karla Borges**[1], **Renata Oliveira Soares**[1],
**Marcelo Buzzi**[2], **Selma Aparecida Souza Kückelhaus**[1] *

**1** Nucleus of Research in Applied Morphology and Immunology, Faculty of Medicine, University of Brasilia, Federal District, Brasilia, Brazil, **2** Innovacorium Inc., Gainesville, Florida, United States of America

\* selmask@gmail.com

**Data Availability Statement:** All relevant data are within the manuscript and its Supporting Information files.

**Funding:** The author(s) received no specific funding for this work.

## Abstract

Blood Concentrates (BCs) are autologous non-transfusional therapeutical preparations with biological properties applied in tissue regeneration. These BCs differ in the preparation method, in fibrin network architecture, growth factors release as well as in platelet/cell content. Methodological changes result in distinct matrices that can compromise their clinical effectiveness. The present study evaluated the influence of different g-forces and types of tubes in the release of vascular endothelial growth factor (VEGF) from platelet-rich fibrin (PRF) as a function of time. The PRF-like samples were obtained with three g-forces (200, 400, and 800 x g) for 10 minutes in pure glass tubes or in polystyrene-clot activator tubes. Scanning and Transmission electron microscopy was used to morphometric analyzes of PRF's specimens and flow cytometry was used to quantify VEGF slow release until 7 days. Our results showed that platelets were intact and adhered to the fibrin network, emitting pseudopods and in degranulation. The fibrin network was rough and twisted with exosomic granulations impregnated on its surface. An increase in the concentration of VEGF in the PRF supernatant was observed until 7 days for all g forces (200, 400 or 800 xg), with the highest concentrations observed with 200 x g, in both tubes, glass or plastic. Morphological analyzes showed a reduction in the diameter of the PRF fibers after 7 days. Our results showed that g-force interferes with the shape of the fibrin network in the PRF, as well as affect the release of VEGF stored into platelets. This finding may be useful in applying PRF to skin lesions, in which the rapid release of growth factors can favor the tissue repair process. Our observations point to a greater clarification on the methodological variations related to obtaining PRF matrices, as they can generate products with different characteristics and degrees of effectiveness in specific applications.

## Introduction

The fibrin matrix is a natural clot that acts as a primordial scaffold for the conduction of tissue repair [1, 2]; it is a polymeric three-dimensional network formed after fibrinogen activation by

**Competing interests:** The authors have declared that no competing interests exist.

enzymatic peptide cleavage [3]. The polymerization occurs instantly by contact with tissue glycoproteins or by thixotropic oxidation in exogenous mechanisms [4].

The biological basis of hemostasis and tissue repair served for the development of obtaining operational methodologies for non-transfusion therapeutic blood concentrates. These methodologies provide a selective concentration of platelets, mononuclear leukocytes, and glycoproteins in the fibrin matrix, elements that are directly related to regenerative responses [5]. These blood concentrates have been used in dentistry and medicine as a surgical adjuvant to provide early tissue repair [6, 7].

Low-speed centrifugation methods generated blood concentrates enriched with platelets from selective blood separation, and methodological advances have provided platelet-rich fibrin (PRF) from blood samples without anticoagulant, which is characterized by elasticity and mechanical resistance and for concentrating mononuclear leukocytes, glycoproteins and growth factors [8, 9].

Currently, platelet concentrates have been the focus of many basic and clinical studies, [10], with the majority of scientific articles mainly addressing its therapeutic potential as a surgical adjuvant in dentistry, such as peri-implantology and maxillofacial surgery [11–14]. Besides, further studies on the treatment of chronic wounds and muscle-tendinous lesions in humans have introduced the topic in the medical field as a promising therapeutic potential [15–21].

The autologous blood concentrates obtained by PRF Standard [22], A-PRF™ [23, 24], Intraspin L-PRF™ [25, 26] CGF™ [27–30], PRGF™ [31, 32], Fibrin System™ [33] and others [34–39] show methodological variations in centrifugation time, g-force, type of rotor and model of the centrifuge, as well as in the types of tubes for blood collection. All together, these methods for obtaining platelet concentrates show differences in the organization of the fibrin matrix and in the release kinetics of growth factors [40–42].

Considering that methodological variations may imply the morphological and biochemical characteristics of PRF, the present study aimed to describe the morphology and the influence of g-force on VEGF release up to seven days after its production.

## Materials and methods

### Individuals and ethical aspects

The participants were healthy, non-smoker, adult women (n = 5) who agreed to donate 120 mL of blood to the study. The present study observed all ethical standards for scientific research with humans in conformity with the Declaration of Helsinki (World Medical Association Recommendation 2013). The Research Ethics Committee approved the study of the Medical School of the University of Brasilia, Brazil, under number 055468/2015.

### Formation of blood concentrates from whole blood

In order, blood samples were obtained pure glass tubes type 10 mL (Montserrat, Brazil) and polystyrene clot activator tubes (Greiner Bio-One, Brazil) by vacuum collection. Samples' blood was collected and immediately transferred to 25° rotor fixed-angle centrifuge Fibrin-Fuge25 (Monserrat, Brazil) centrifugated with 200, 400, and 800 x g at 10 minutes.

### Ultrastructural analyses by scanning electron microscopy

PRF clot's fragment obtained by 200, 400, and 800 x g / 10 minutes were separated and conditioned in plastic tubes free of additives (Greiner Bio-One, Brazil). Immediately after PRF production and after 7, 14 or 21 days at 37 °C, the specimens were sectioned (body, buffy coat and proximal sediment) and fixed in 2% glutaraldehyde solution and 2% paraformaldehyde in

0.1M sodium cacodylate buffer, pH 7.2. After being fixed, they were washed in 0.1M sodium cacodylate buffer, post-fixed in osmium tetroxide (1%) in sodium cacodylate buffer, then the specimens were dehydrated for 15 minutes in solutions with increasing concentrations of acetone (30%, 50%, 70%, 90% and 3x 100%). Then, the specimens were dried to the critical point with $CO_2$ and metalized with gold. For analyzes by scanning electron microscopy (JEOL 7001F, Tokyo, Japan), the images (20 to 20000 x of magnification) were analyzed using GIMP 2.10 software (Cockroach Labs, USA). To the morphometric study we evaluated the diameter of twenty fibers using the digital planimeter of Autocad software (Autodesk, USA).

## Ultrastructural analyses by transmission electron microscopy

For transmission electron microscopy, analyzes PRF fragments in the buffy coat region were fixed and processed as previously reported. After being impregnated with PolyBed 812 resin (PolySciences, USA) the fragments were sectioned on an ultramicrotome to obtain the images (Jeol 100CXII, Japan). Qualitative analyzes where performed in digital images using the GIMP 2.10 software (Cockroach Labs, USA).

## VEGF quantification

The PRF of individuals were maintained up to 7 days at 37˚C and after 0, 1, 2, 3, 4, 5, 6 or 7 days an aliquot of the serum (200 μL) was cryopreserved (-80˚ C) until analyzes. Quantification of VEGF was performed using the bead array (CBA) cytometry method, used in the commercial kit (VEGF Flex Set human kit and BD ™ CBA cell signaling main kit, USA), according to the manufacturer's specifications on the flow cytometer LSR Fortessa ™ (BD ™, USA). FCAP Array ™ software version 3.0 (BD ™, USA) was used to calculate the concentration of VEGF based on a curve of the pattern of this growth factor. Results were expressed in pg/mL.

## Statistical analyzes

The statistical analysis used Bartlett's test for equal variances and the Kolmogorov–Smirnov test for normal distribution before comparative tests. The analyzes were performed by the pared t test or Wilcoxon to compare samples of normally or non-normally distributed data, respectively, and linear regression. The GraphPadPrism 8.0 software package (GraphPad, USA) was employed for statistical tests and graphical presentation of the data. Differences with a two-tailed value of $p < 0.05$ were considered statistically significant.

## Results

### Morphological analyzes

The qualitative analyzes represented in Fig 1, showed that the PRF (Fig 1A) is divided into three distinct regions, the upper part being formed by a fibrin network (B), the intermediate part (medium body) containing platelets immersed in the fibrin network (C) and the lower part (buffy coat) containing the largest fraction of platelets and leukocytes (D).

The platelets were intact and adhered to the fibrin network, emitting pseudopods and in degranulation, as shown in Fig 1E, 1F, 1G and 1H. On the surface of the platelets were observed rough and agglomerated bodies with dimensions of 3.0 ± 2.0 μm. Leukocytes were closely associated with platelets and presented a rough surface. The fibrin network exhibited variations in its three-dimensional organization, were rough and twisted, in addition to exhibiting polymeric chains with exosomatic granulations impregnated on the surface (Fig 1I and 1J).

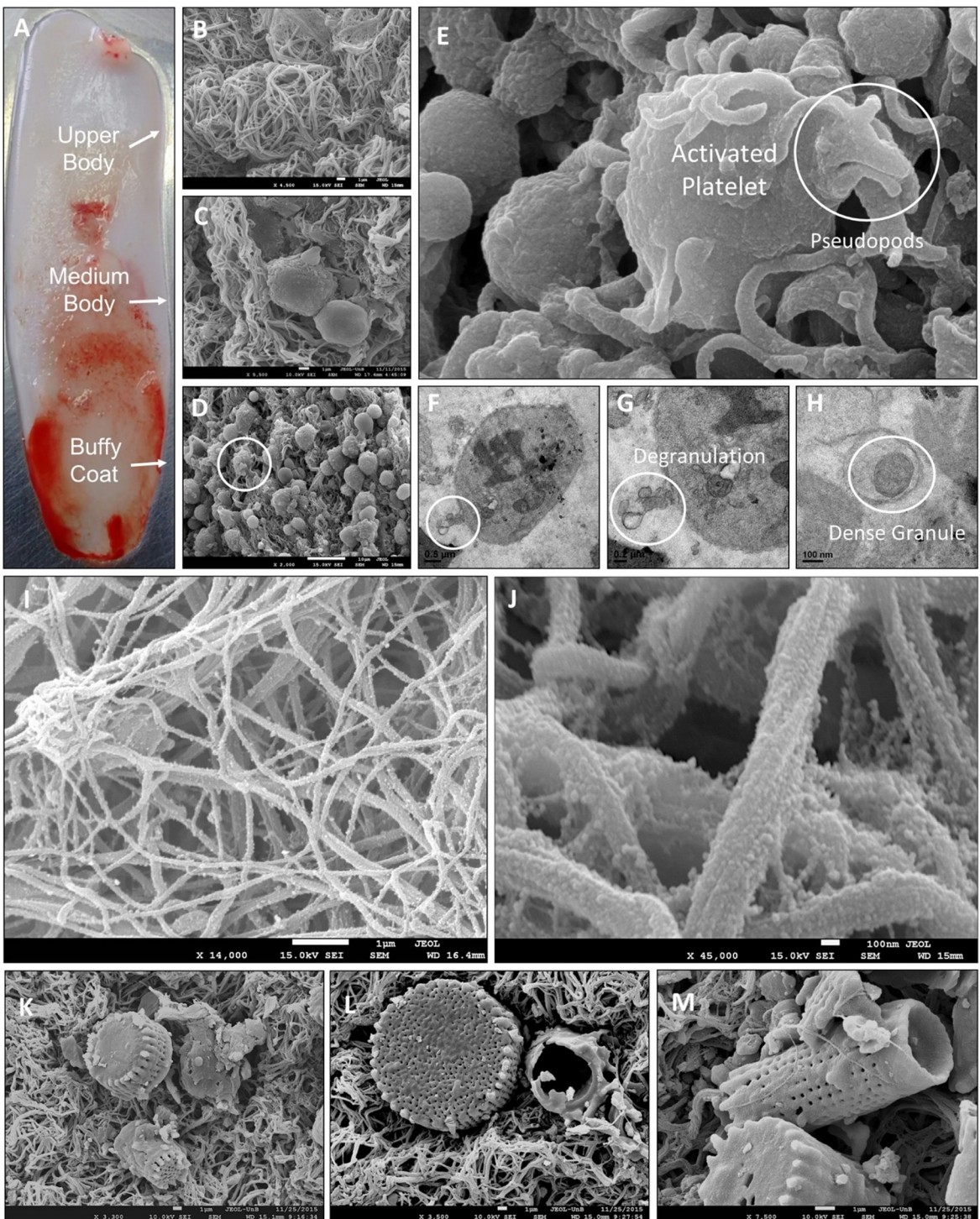

**Fig 1.** Photography showing in A the macroscopy of three regions of PRF (upper body, medium body and buffy coat) and eletrocmicrographs (B to M). Observe the ultrastructural arrangements of the three regions of PRF (B, C, D), activated platelet (E) with pseudopods and degranulation process (F, G, H). Also, observe the fibrin surface with polymeric chains and impregnated superficial nanoparticles (I, J). In K, L, M are shown interspersed plankton diatoms frustules in PRF matrix obtained by Vacuette™ plastic tube with a clot activator.

Fig 1K, 1L and 1M show the PRF obtained with the $SiO_2$ platelet activator in which the planktonic diatoms frustoles are interspersed in the matrix.

## Quantification of VEGF

Our results showed an increase in the concentration of VEGF in the PRF supernatant over the 7 days of evaluation and for all g forces (200, 400 or 800 x g), with the highest concentrations observed with 200 x g, in both tubes, glass (linear regression, $r^2 > 0.96$, $p < 0.0001$) or plastic (linear regression, $r^2 > 0.86$, $p < 0.001$) (Fig 2A and 2B). However, the morphometric analyzes showed a reduction in the diameter of the PRF fibers after 7 days at 37° C (Fig 2C and 2D) and qualitatively, these analyzes indicated a decrease in the density of the fibrin network after the 7 days (Fig 2E2, 2F2 and 2G).

## Discussion

The study of fibrinolysis has two strands of high relevance. Within the vessels, this physiological process quickly prevents thromboembolism, while in the extravascular environment this process is slow because fibrin acts structurally in the injured tissue [43, 44]. Considering the application of PRF in tissue repair, the elucidation of how fibrinolysis occurs in the extravascular environment can favor its therapeutic application.

For the investigation of fibrinolysis, qualitative and quantitative methods of high relevance in the clinical sphere are available to differentiate pathological conditions and indicate the appropriate therapy, which commonly involve anti-fibrinolytics, such as $\alpha_2$-antiplasmin and $\alpha_2$-macroglobulin [45–49]. In this study, the objective was to understand the kinetic of fibrinolysis obtained with different g forces in vitro.

Our results showed that the variation in g-force for the production of PRF interferes with the shape of the fibrin network and in the VEGF release. Considering that the platelet granules were attached on the surface of the fibrin fibers, as shown in the images obtained by SEM, it is speculated that the lower g force applied (200 x g) is sufficient to promote the platelet activation and the exosomes release. Furthermore, the lower g-force promote highest concentration of VEGF and decrease of the fibrin fibers diameter. This finding may be useful in applying PRF to skin lesions, in which the rapid release of growth factors can favor the tissue repair process. Besides, this fibrin network can be used as a drug system delivery [50, 51], acting to different therapeutic applications.

It has been reported that the slow release of growth factors from the PRF is fundamental to support the therapeutic application of this platelet concentrate [52, 53]. Other studies have shown that the concentration of growth factors varies according to the production protocol of PRF [54, 55], mainly related to the g-force and the type of tube used.

Our results demonstrated that the use of the glass tube provided higher concentrations of VEGF released in all g-force bands, especially after 72 hours, when compared to those obtained in plastic tubes with clot activator. Bonazza et al., 2016 [29] when comparing the effects of three different types of tubes in the PRF matrix, obtained by the CGF™ method, showed difference in the platelets and leukocytes dispersion and the fibrin density mesh observable by immunohistochemical assays. The fibrinolysis experiment demonstrated that the concentration of growth factors increases as the fibrin fibers reduce their thickness and demonstrate detachment of particles from their surface. Thus, the lower concentration of VEGF in the exudate can have two meanings; 1) the peptide was degraded early; 2) the peptide is firmly adhered and protected in the fibrin matrix bed.

The present experiment reproduced the immunoassay methodologies reported in the literature and served to demonstrate that PRF matrices have supraphysiological concentrations of

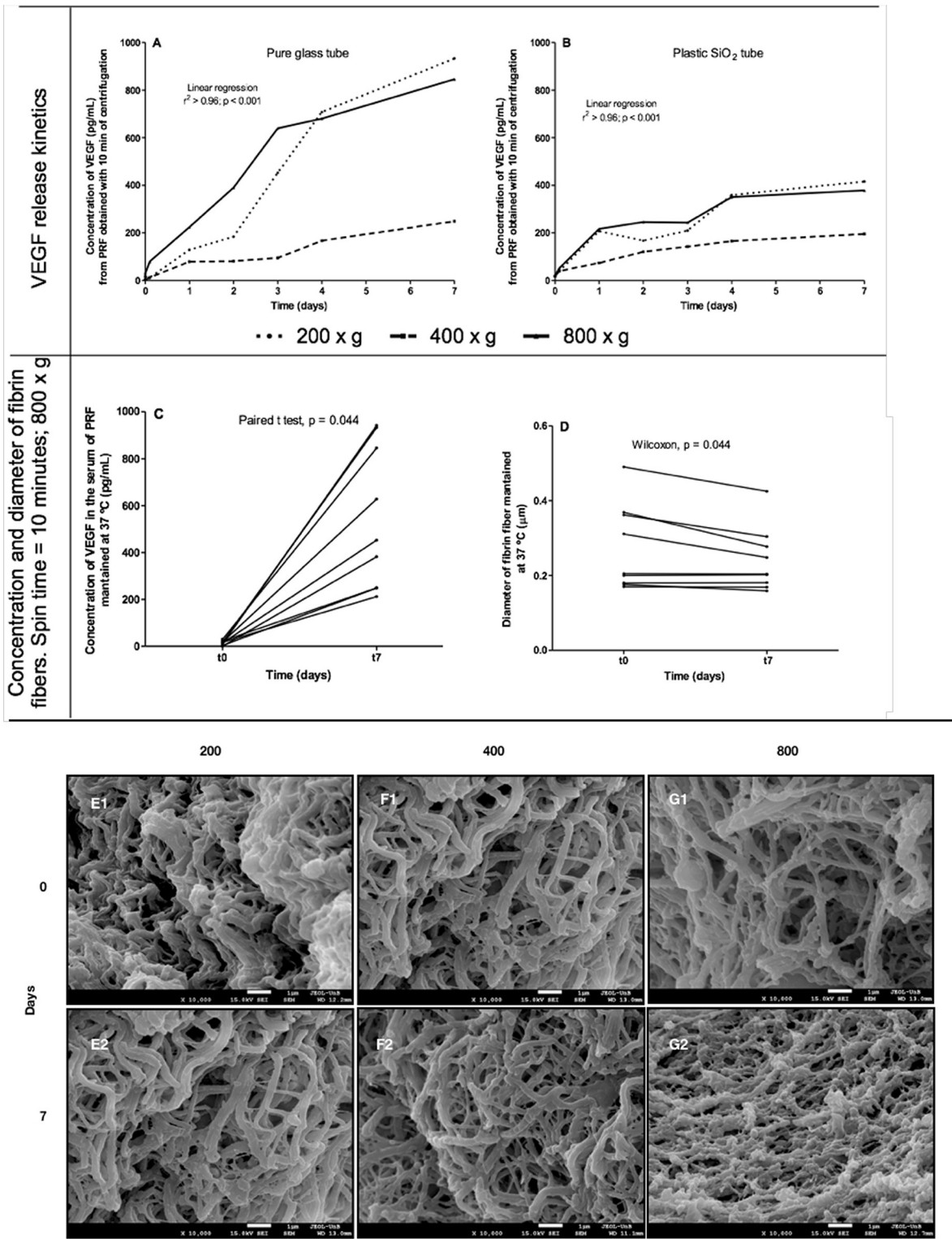

**Fig 2.** Kinetics of VEGF release from the fibrin network to the PRF supernatant obtained in a glass (A) or plastic (B) tube over 7 days. In C and D are observed the paired analyzes of the total VEGF concentration (C) and the diameter of the fibrin fibers (D). In E, F, G are observed the fibrin network images obtained with 200 x g (E), 400 x g (F) or 800 x g (G) at t0 and t7, indicating a decrease in the density of the fibrin network after 7 days. (E2, F2, G2). The results showed greater VEGF release (linear regression) and smaller diameter of fibrin fibers after 7 days of PRF at 37°C (paired analyzes).

growth factors. In vitro experiments are very limited since they are performed under controlled environmental and metabolic conditions, unlike *in vivo* behavior.

Activated platelets and leukocytes release growth factors in the medium where they are inserted. In this way, SEM in large increases was able to demonstrate the impregnation of nanometric particles attached to fibrin fibers. Dohan, et al., 2010 [7] presented the concept of the three-dimensional model of the fibrin matrix that accommodates growth factors by adhesion. In Fig 2, we identified this configuration in a real model by SEM micrographs.

The relationship between the PRF's morphology and the VEGF release showed the interdependence of these processes. However, the released concentration, by itself, does not constitute any intrinsic advantage of the method of obtaining since the cell signaling resulting from growth factors is performed in the three-dimensional bed itself. The highest concentrations observed in vitro occurred precisely in the period of fibrinolysis [48]. It is also worth considering that these concentrations released from the matrix in vitro do not faithfully reproduce what occurs in tissue beds in vivo, where the action of enzymes, oxidizing agents, and migratory cells interfere with the physiological activity of these growth factors. Even because the fibrin matrix exerts a proteolytic protection mechanism on these proteins, keeping them functionally adhered for the cellular signaling that is fundamental during tissue repair [49].

Another important microscopic observation was the impregnation of different particles of silica in the PRF matrix obtained in plastic tubes with clot activators. Tubes from commercial brands presented silicon extracts from algalic sources, one of them being a composition of diatom shells. Clot activators act as oxidizing agents for in vitro diagnostic purposes [56–58].

In Vitro Devices (IVDs), despite being sterile to ensure the reliability of laboratory tests, they have limitations for medical use. According to international health standards, processed products, including blood and blood products, as well as handling devices, must have a Medical Device (MD) classification [8]. In Europe, the EU regulation 2017/745 of the European Parliament and the Council already recommends the disuse of tubes manufactured for laboratory purposes and the adoption of devices with MD classification.

The company Silfradent in Santa Sofia, Italy, is one of the pioneers in this adaptation and already has these devices with such classification. However, despite this international sanitary adequacy, there have been no reports of cytotoxicity or therapeutic failures in the literature over the past 20 years that could be associated with the use of IVD tubes [57, 59, 60]. Tsujino et al., 2019 [58] indicated potential risks for the use of tubes containing amorphous silica as a clot activator [52], and Kawase et al., 2020 [60] demonstrated cytotoxicity in periosteal cells by contact with the silica impregnated in the tubes. There is a strong trend towards standardization for the use of tubes that are free of chemical additives.

The manufacture of plastic tubes with clot activators supplied a demand for biosafety in the field of laboratory diagnosis [61]. The glass tubes serve to accelerate the clot retraction; however, they are susceptible to breakage during handling and centrifugation, increasing the risk of accidents at work. In this way, the use of plastic tubes with clot activators, in laboratory logistics, accelerate the formation of the clot and reduce the risks of these accidents [4].

Bowen and Remaley, 2014 [62] clarified that laboratory analytical methods are susceptible to variations in additives used in the manufacture of tubes for in vitro diagnostics, thus interfering with the results of the analyzes. Contaminating agents such as lead and other heavy metals present in the rubbers and lubricants used to close the tubes can affect the accuracy of laboratory tests.

Some companies involved in the sale of centrifuges and supplies for obtaining blood concentrates, in compliance with international health regulations, already have in their sterile portfolio tubes in specific packages with the indicative description for in vivo use [28].

Despite this international health trend regarding the indication of equipment and supplies classified as a medical device (MD), there is no indication in the international literature of

inefficiency, toxic or immunogenic implications due to the use of IVD tubes containing clot activators in obtaining PRF matrices.

Due to the glass tube presented the better kinetic pattern in the slow release of VEGF, it can be indicated for obtaining the PRF matrices for non-transfusion therapeutic use with safety. Studies using experimental models in vivo and in vitro are necessary to clarify the cytotoxic potential of these activators.

## Conclusion

The microscopic and flow cytometric experiments showed the effect of the different experimental conditions, where the smallest g-forces were more promising concerning cell composition and VEGF release. Our results showed that g-force interferes with the shape of the fibrin network in the PRF, as well as affect the VEGF release. This finding may be useful in applying PRF to skin lesions, in which the rapid release of growth factors can favor the tissue repair process. Our observations point to a greater clarification on the methodological variations related to obtaining PRF matrices.

## Acknowledgments

We are grateful to technical staffs Biology Institute's microscopy and microanalysis laboratory —University of Brasilia for supporting.

## Author Contributions

**Conceptualization:** Leonel Alves de Oliveira, Selma Aparecida Souza Kückelhaus.

**Data curation:** Leonel Alves de Oliveira, Selma Aparecida Souza Kückelhaus.

**Formal analysis:** Leonel Alves de Oliveira, Selma Aparecida Souza Kückelhaus.

**Funding acquisition:** Leonel Alves de Oliveira, Selma Aparecida Souza Kückelhaus.

**Investigation:** Leonel Alves de Oliveira, Tatiana Karla Borges, Selma Aparecida Souza Kückelhaus.

**Methodology:** Leonel Alves de Oliveira, Tatiana Karla Borges.

**Project administration:** Leonel Alves de Oliveira.

**Resources:** Leonel Alves de Oliveira, Marcelo Buzzi, Selma Aparecida Souza Kückelhaus.

**Software:** Leonel Alves de Oliveira, Selma Aparecida Souza Kückelhaus.

**Supervision:** Marcelo Buzzi, Selma Aparecida Souza Kückelhaus.

**Validation:** Leonel Alves de Oliveira.

**Visualization:** Leonel Alves de Oliveira, Selma Aparecida Souza Kückelhaus.

**Writing – original draft:** Leonel Alves de Oliveira, Selma Aparecida Souza Kückelhaus.

**Writing – review & editing:** Leonel Alves de Oliveira, Renata Oliveira Soares, Selma Aparecida Souza Kückelhaus.

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
