## [Decision Letter · Decision Letter 0]

13 Jul 2020

PONE-D-20-14559

Methodological variations affect the release of VEGF in vitro and fibrinolysis' time from platelet concentrates

PLOS ONE

Dear Dr. Kückelhaus,

Thank you for submitting your manuscript to PLOS ONE. After careful consideration, we feel that it has merit but does not fully meet PLOS ONE’s publication criteria as it currently stands. Therefore, we invite you to submit a revised version of the manuscript that addresses the points raised during the review process.

We look forward to receiving your revised manuscript.

Kind regards,

Amitava Mukherjee, ME, Ph.D.

Academic Editor

PLOS ONE

Journal Requirements:

2. Thank you for including your competing intersests statement; "The authors have declared that no competing interests exist."

We note that one or more of the authors are employed by a commercial company:

Innovacorium Inc

Reviewers' comments:

Reviewer's Responses to Questions

**Comments to the Author**

1. Is the manuscript technically sound, and do the data support the conclusions?

Reviewer #1: Yes

2. Has the statistical analysis been performed appropriately and rigorously? 

Reviewer #1: Yes

3. Have the authors made all data underlying the findings in their manuscript fully available?

Reviewer #1: Yes

4. Is the manuscript presented in an intelligible fashion and written in standard English?

Reviewer #1: No

5. Review Comments to the Author

Reviewer #1: This paper describes the formation of platelet rich fibrin using different G forces and different tube materials. Outcomes are described as fibrin appearance (using EM) and VEGF formation (CBA) in supernatant. The investigators measured VEGF formation as indicator of quality of PRF formed. It would be useful to measure other biochemical indicators such as PDGF, markers on platelets in the matrix. Additionally, the FBE of the blood from which the PRF was formed should also be reported either in the manuscript or in the supplementary. The language in the paper is difficult to read at times (e.g. parameters = methods, obtention of blood-derived concentrates by peripheral venipuncture and centrifugation = formation of blood concentrates from whole blood).

6. PLOS authors have the option to publish the peer review history of their article (what does this mean?). If published, this will include your full peer review and any attached files.

Reviewer #1: No

---

## [Author Response · Author response to Decision Letter 0]

8 Sep 2020

Response to Reviewer #1: 

Considering the reviewer's suggestion about measuring other biochemical indicators, we highlight that it is no longer possible to perform new experiments, since the samples were discarded after the VEGF analyzes. However, we emphasize that the suggestion will be considered in future experiments.

Regarding the revision of the language, we inform that grammatical changes were made in the ressubmission, in order to improve the text.

---

## [Decision Letter · Decision Letter 1]

21 Sep 2020

Methodological variations affect the release of VEGF in vitro and fibrinolysis' time from platelet concentrates

PONE-D-20-14559R1

Dear Dr. Kückelhaus,

We’re pleased to inform you that your manuscript has been judged scientifically suitable for publication and will be formally accepted for publication once it meets all outstanding technical requirements.

Kind regards,

Amitava Mukherjee, ME, Ph.D.

Academic Editor

PLOS ONE

Additional Editor Comments (optional):

Reviewers' comments:

Reviewer's Responses to Questions

**Comments to the Author**

1. If the authors have adequately addressed your comments raised in a previous round of review and you feel that this manuscript is now acceptable for publication, you may indicate that here to bypass the “Comments to the Author” section, enter your conflict of interest statement in the “Confidential to Editor” section, and submit your "Accept" recommendation.

Reviewer #1: All comments have been addressed

2. Is the manuscript technically sound, and do the data support the conclusions?

Reviewer #1: Yes

3. Has the statistical analysis been performed appropriately and rigorously? 

Reviewer #1: Yes

4. Have the authors made all data underlying the findings in their manuscript fully available?

Reviewer #1: Yes

5. Is the manuscript presented in an intelligible fashion and written in standard English?

Reviewer #1: Yes

6. Review Comments to the Author

Reviewer #1: (No Response)

7. PLOS authors have the option to publish the peer review history of their article (what does this mean?). If published, this will include your full peer review and any attached files.

Reviewer #1: No

---

## [Editor Report · Acceptance letter]

24 Sep 2020

PONE-D-20-14559R1 

Methodological variations affect the release of VEGF * in vitro* and fibrinolysis' time from platelet concentrates 

Dear Dr. Kückelhaus:

I'm pleased to inform you that your manuscript has been deemed suitable for publication in PLOS ONE. Congratulations! Your manuscript is now with our production department. 

Kind regards, 

on behalf of

Professor Dr. Amitava Mukherjee 

Academic Editor

PLOS ONE